# Corticonigral projections recruit substantia nigra pars lateralis dopaminergic neurons for auditory threat memories

Lorenzo Sansalone[1,2], Emily L. Twedell [1,2,3], Rebekah C. Evans[1,4], Alejandra Boronat-Garcia[1,2], Renshu Zhang[1,2] & Zayd M. Khaliq [1,2] ✉

Dopaminergic neurons (DANs) in the substantia nigra pars lateralis (SNL) project to the tail of striatum, where they contribute to threat behaviors. Auditory cortex contributes to threat conditioning, but whether it directly modulates DANs is unclear. Here, we show that SNL DANs fire irregularly, achieve rapid maximal firing rates, exhibit distinct ionic conductances, and receive predominantly excitatory input. This contrasts with substantia nigra pars compacta (SNc) DANs that fire regularly and receive mainly inhibitory input, establishing SNL DANs as a physiologically distinct dopaminergic subpopulation. Functional mapping revealed robust excitatory input from auditory and temporal association cortices to SNL DANs, but not SNc DANs. In behavioral experiments, inhibiting neurotransmitter release from either SNL DANs or cortical afferents to SNL resulted in impaired auditory threat conditioning. Thus, our work reveals robust functional corticonigral projections to SNL DANs which directly regulate threat behaviors.

Midbrain dopaminergic neurons (DANs), traditionally linked to reward processing[1,2], have been increasingly recognized for their role in aversive and threat behaviors[3–6]. Dopamine signaling in regions associated with threat processing, such as the amygdala[7–9], prefrontal cortex[10,11], and striatum[12–15], has been well documented. However, there has been comparatively less investigation into the involvement of specific dopaminergic neuron subpopulations and the circuit projections driving their activity during threat behaviors.

Substantia nigra pars lateralis (SNL) DANs project to the tail of the striatum, where they play a crucial role in processing salient, novel and aversive stimuli[16–21], yet little is understood about how the surrounding circuits modulate their activity during threat behaviors. Optogenetic stimulation of dopaminergic axons within the tail of the striatum causes avoidance of novel objects[12] while inhibition reduces auditory fear responses[15]. Chemical lesioning of these projections to the tail of the striatum reduces avoidance behaviors, suggesting their importance for threat avoidance[12].

The activity of SNL DANs is predominantly regulated by inputs from the central nucleus of the amygdala, that act on SNL GABAergic neurons to produce disinhibition of SNL DANs[22]. Moreover, anatomical studies using viral-genetic mapping have identified projections to substantia nigra DANs, showing that subthalamic nucleus provides particularly selective input to DANs projecting to the tail of the striatum[16]. However, the excitatory projections that drive SNL DANs during threat behaviors are not yet known.

Auditory threat conditioning[23] engages the auditory cortex, which then interacts with the amygdala[24] and striatum[25,26] to enable acquisition and establishment of auditory threat memories. However, whether cortical inputs engage DANs during threat signal processing through a direct corticonigral pathway remains unknown. Anatomical studies have identified cortical projections to substantia nigra DANs[16,27–29], while functional results suggest that corticonigral projections to SNc DANs from motor cortex are sparse[30]. Whether auditory cortex provides direct projections to midbrain DANs located in SNc or

[1]Cellular Neurophysiology Section, National Institute of Neurological Disorders and Stroke Intramural Research Program, National Institutes of Health, Bethesda, MD, USA. [2]Aligning Science Across Parkinson's (ASAP) Collaborative Research Network, Chevy Chase, MD, USA. [3]Present address: Macalester College, Saint Paul, MN, USA. [4]Present address: Georgetown University Medical Center, Washington, DC, USA. ✉e-mail: zayd.khaliq@nih.gov

SNL, and whether these projections contribute to threat conditioning, have not yet been determined.

Here, we demonstrate that the auditory association cortex, comprising secondary auditory cortex (AuV) and temporal association cortex (TeA), provides specific projections to SNL DANs that contribute significantly to memory retrieval during auditory threat conditioning. We show that optical activation of projections from the auditory cortex robustly increases firing in SNL DANs. By contrast, we found that lateral or medial SNc DANs (lSNc or mSNc) receive virtually no functional input from auditory association cortex. Comparative analysis of the firing properties of SNL, lateral SNc, and medial SNc DANs showed that SNL DANs represent an independent neuronal subpopulation characterized by irregular pacemaking and significantly higher maximal firing rates. These findings are further supported by biophysical experiments demonstrating differences in underlying ion channels expressed in SNL and SNc DANs. Finally, auditory threat conditioning experiments showed that disrupting synaptic transmission from SNL DANs to the tail of the striatum using virally-expressed tetanus toxin significantly reduced threat learning (CS-US association) during Pavlovian paradigms. Importantly, preventing synaptic release from the auditory cortex to SNL DANs affected both threat learning and retrieval of auditory threat memories. Taken together, our findings reveal a corticonigral pathway that directly influences the activity of midbrain DANs involved in auditory-related aversive behaviors. These results offer insights into how cortical-midbrain interactions influence dopaminergic transmission and contribute to threat behaviors. These findings may enhance our understanding of the altered sensory processing associated with post-traumatic stress disorder (PTSD) and phobias[31].

## Results

### SNL DA neurons exhibit distinct intrinsic firing properties relative to SNc DA neurons

Dopaminergic neurons (DANs) in both the substantia nigra pars lateralis (SNL) and pars compacta (SNc) neurons project to the tail of the striatum (TS), but whether SNL and SNc DANs differ in their morphology and physiology has not yet been determined. To identify SNL DANs, we first performed in situ hybridization experiments to detect cells co-expressing tyrosine hydroxylase (TH) with either calbindin or glutamate vesicular transporter 2 (VGluT2)[32–36].

Calbindin-positive (Calb + ) and VGlutT2-positive (VGlutT2 + ) DANs were present in the SNL and SNc (Supplementary Fig. 1 and 4), consistent with previous findings[34]. Moreover, Calb+ DANs almost always co-expressed VGluT2 (Supplementary Fig. 2). As an alternative strategy to identify SNL DANs, we employed an intersectional genetic approach by crossing either Calb- Cre or VGluT2-Cre mice to DAT-Flp::Ai65 mice (Supplementary Fig. 3). This approach similarly revealed Calb+ and VGluT2+ DANs in both the SNL and SNc (Supplementary Fig. 4), raising the question of whether these neurons comprise distinct subpopulations.

To compare DANs in the SNL and SNc, we first quantified the size of neurons by taking the cross-sectional somatic area of TH-positive neurons using in situ hybridization in fixed midbrain slices from C57BL/6 J mice (Fig. 1a–c). SNc DANs had large cell bodies with no significant difference between medial SNc (mSNc) and lateral SNc (lSNc) neurons (somatic area; mSNc, $402.98 \pm 11.75\ \mu m^2$, $n = 130$; lSNc, $413.63 \pm 20.28\ \mu m^2$, $n = 53$, $p = 0.78$). However, we found that SNL neurons were ~30% smaller with an average somatic area of $287.71 \pm 12.72\ \mu m^2$ (SNL: $n = 47$; SNL vs. lSNc, $p = 1.07 \times 10^{-5}$; SNL vs. mSNc, $p = 2.61 \times 10^{-10}$). Importantly, results from genetic reporter lines (Calb-Cre or VGluT2-Cre DAT-Flp Ai65) were similar, with average somatic area of Calb+ and VGluT2+ SNL DANs being up to 60% smaller than SNc DANs (Supplementary Fig. 4).

We next performed whole-cell patch-clamp electrophysiology experiments to examine the intrinsic firing properties of SNL DANs in Calb- Cre or VGluT2-Cre DAT-Flp::Ai65 mice. We found that SNL DANs fire spontaneously at significantly lower rates compared to SNc DANs (Fig. 1d–g) (avg firing rate; SNL, $1.15 \pm 0.13$ Hz, $n = 53$; mSNc, $2.80 \pm 0.13$ Hz, $n = 23$; lSNc, $2.90 \pm 0.32$ Hz, $n = 11$; SNL vs. mSNc, $p = 1.08 \times 10^{-10}$; SNL vs. lSNc, $p = 2.07 \times 10^{-6}$). Pacemaking in SNc DANs was highly rhythmic[37]. By contrast, we found that pacemaking in SNL DANs is highly irregular, with an average coefficient of variation of interspike interval (CV ISI) of $43.14 \pm 4.26\%$ ($n = 48$) compared to SNc DANs, which display CV ISI lower than 10% (Fig. 1h; CV ISI; mSNc, $3.94 \pm 0.42\%$, $n = 23$; lSNc, $6.37 \pm 0.75\%$, $n = 11$; SNL vs. mSNc, $p = 7.55 \times 10^{-19}$; SNL vs. lSNc, $p = 7.15 \times 10^{-12}$).

Similar observations were made with cell-attached and perforated-patch electrophysiology experiments, suggesting that the irregular firing in SNL DANs did not reflect wash-out of intracellular signaling proteins in whole-cell recordings (Supplementary Fig. 5). Moreover, CV ISI increased across the mediolateral axis of the substantia nigra of DAT-Cre Ai9 mice (Fig. 1i), from mSNc to SNL. Additionally, we found an inverse relationship between CV ISI and the capacitance of DANs (Supplementary Fig. 7), indicating that the observation of higher CV ISI correlates with smaller neuron sizes. Comparison of firing in VGluT2-Cre mice revealed that SNL DANs differ substantially from those located in lSNc (Supplementary Fig. 6). Testing maximal firing rates with direct current-injections (Fig. 1j–m) revealed that SNc neurons exhibited low maximal firing rates (average maximal firing rate, mSNc, $20.93 \pm 0.78$ Hz, $n = 13$; lSNc, $16.12 \pm 1.04$ Hz, $n = 4$) consistent with past work[38]. By contrast, we found that SNL DANs exhibited substantially higher average maximal firing rates of $45.08 \pm 4.85$ Hz ($n = 15$; SNL vs. mSNc, $p = 5.34 \times 10^{-4}$; SNL vs. lSNc, $p = 5.16 \times 10^{-4}$). Thus, comparison of SNL and SNc DANs intrinsic firing properties suggests that they represent two different dopaminergic neuron subpopulations.

When examining the interspike voltage in SNL DANs during whole-cell patch-clamp recordings, we noticed the presence of subthreshold post-synaptic potentials (PSPs). The bulk of these PSPs were abolished by CNQX and D-AP5, inhibitors of excitatory synaptic AMPA and NMDA receptors (Fig. 1n and Supplementary Fig. 10). Importantly, inhibition of synaptic transmission had no significant effect on the firing rate (Fig. 1o; control vs. AP5/NBQX, $p = 0.65$; control vs. AP5/NBQX/GBZ/CGP, $p = 0.89$) or CV ISI (Fig. 1p, control vs. AP5/NBQX, $p = 0.43$; SNL

AP5/NBQX/GBZ/CGP, $p = 0.44$), suggesting that the irregular pacemaking results from the intrinsic properties of SNL DANs. We found that SNL DANs have more depolarized AHP voltages during spontaneous pacemaking compared to SNc DANs (Supplementary Fig. 8a, b), which is further supported by the presence of lower SK-conductances (Supplementary Fig. 8c, d). We also recorded smaller sag voltages during direct hyperpolarization in SNL DANs relative to SNc DANs (Supplementary Fig. 9a, b), indicative of weaker hyperpolarization-activated cyclic nucleotide-gated (HCN) channels recruitment (Supplementary Fig. 9c–e). To more accurately quantify synaptic transmission, we performed voltage-clamp recordings and compared miniature excitatory and inhibitory post-synaptic currents (mEPSC and mIPSC) in SNL and SNc DANs (Fig. 1q). We found that SNL DANs exhibit substantially higher mEPSC frequency with an average of $14.32 \pm 3.47$ Hz compared to $1.26 \pm 0.31$ Hz for SNc neurons (Fig. 1r, SNL: $n = 8$; SNc, $n = 8$, $p = 3.11 \times 10^{-4}$). By contrast, the frequency of mIPSCs was significantly lower in SNL DANs compared to SNc DANs (Fig. 1s; SNL, $0.82 \pm 0.14$ Hz, $n = 6$; SNc, $2.37 \pm 0.34$ Hz, $n = 8$, $p = 0.012$). Therefore, our analysis of miniature events shows that SNL DANs exhibit a high ratio of excitatory to inhibitory (E/I) events, suggesting that they receive primarily excitatory input, which differs substantially from SNc DANs, which are known to be governed mainly by inhibition[39].

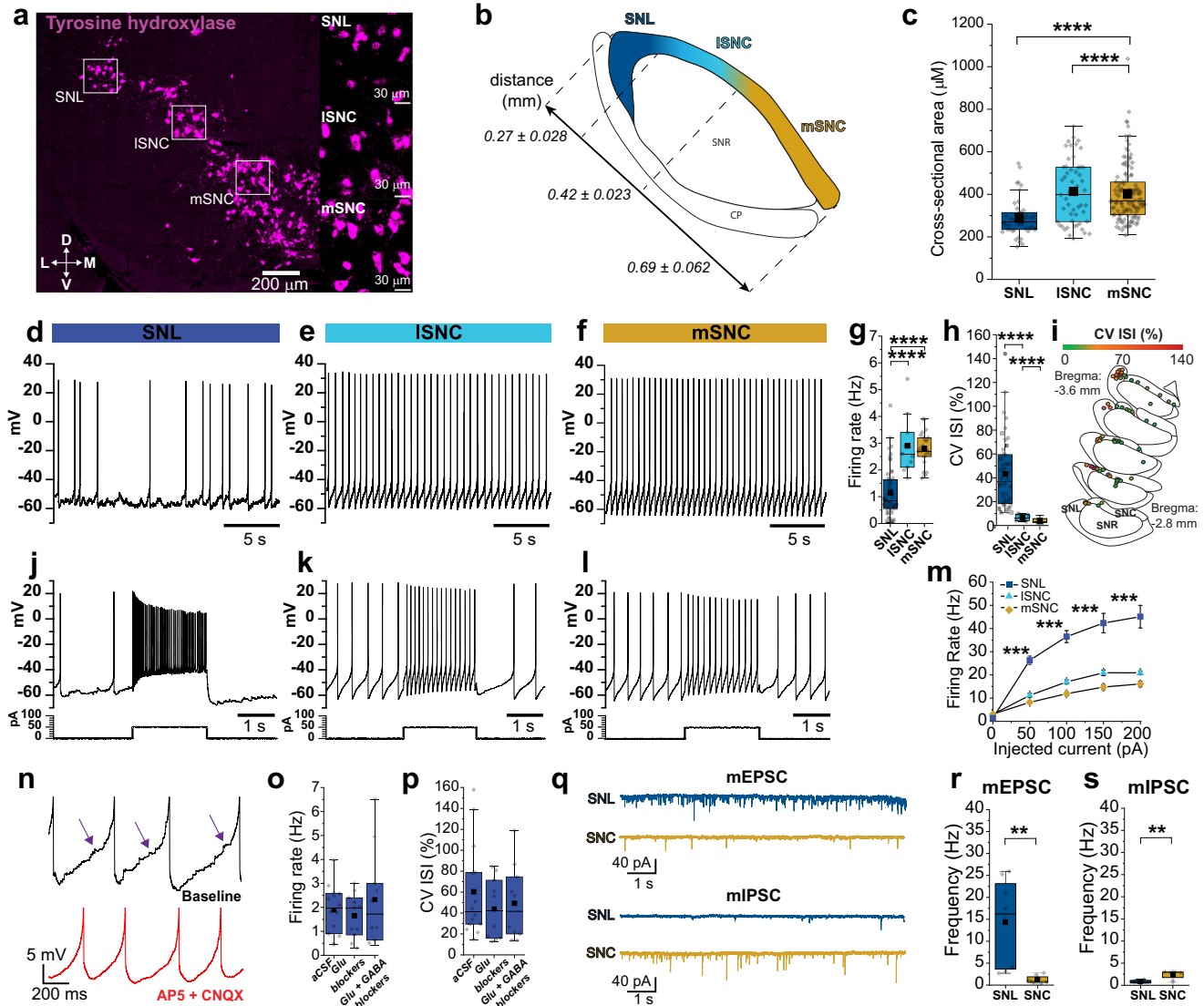

**Fig. 1 | Distinct intrinsic firing and synaptic properties in SNL DA neurons. a** In situ hybridization of TH+ neurons from coronal midbrain sections. *Insets*, SNL (*top*), lateral SNc (lSNc, *middle*), medial SNc (mSNc, *bottom*). **b** Illustrative map of nigral divisions. Numerical values represent average length of subdivision in mm ± s.e.m. **c** Bar plots of somatic cross-sectional areas from TH+ neurons in SNL (*n* = 47), lSNc (*n* = 53), and mSNc (*n* = 130) (SNL vs. lSNc, *p* = 1.07 × 10⁻⁵; SNL vs. mSNc, *p* = 2.61 × 10⁻¹⁰, *N* = 4 mice, two-sided unpaired Mann-Whitney). **d–f** Representative whole-cell recordings of DANs spontaneous firing. **g** Firing rate averages in DANs in SNL (*n* = 53, *N* = 12 mice), lSNc (*n* = 11, *N* = 5 mice), and mSNc, (*n* = 23, *N* = 15 mice) (SNL vs. mSNc, *p* = 1.08 × 10⁻¹⁰; SNL vs. lSNc, *p* = 2.07 × 10⁻⁶, two-sided unpaired Mann-Whitney). **h** CV ISI averages in DANs in SNL (*n* = 53, *N* = 12 mice), lSNc (*n* = 11, *N* = 5 mice), and mSNc (*n* = 23, *N* = 15 mice) (SNL vs. lSNc, *p* = 7.15 × 10⁻¹²; SNL vs. mSNc, *p* = 7.55 × 10⁻¹⁹, two-sided unpaired Mann-Whitney). **i** Plots of cell location within the SN mediolateral axis with neurons color-coded according to CV ISI. **j–l** Representative recordings of evoked firing in DANs (same cells as above). **m** Frequency-current averages in DANs in SNL (*n* = 15, *N* = 6 mice), lSNc (*n* = 4, *N* = 3

mice), and mSNc (*n* = 13, *N* = 8 mice) (SNL vs. lSNc, *p* = 5.16 × 10⁻⁴; SNL vs. mSNc, *p* = 5.34 × 10⁻⁴, two-sided unpaired Mann-Whitney). Data are presented as mean ± SEM. **n** *Top*, Representative trace of post-synaptic potentials (PSPs) during interspike voltage (*arrows*). *Bottom*, Trace from the same neuron following bath application of AP5/CNQX. Note suppression of PSPs. **o, p** Firing rate and CV ISI averages from SNL DANs in control (*n* = 14), plus glutamatergic receptor antagonists (AP5/NBQX, *n* = 11), and plus GABA antagonists (AP5/NBQX/GBZ/CGP, *n* = 10) from *N* = 5 mice. **q** Representative mEPSCs (*top*) and mIPSC (*bottom*) from SNL and SNc DANs recorded in the presence of TTX plus gabazine (mEPSC) or AP5-NBQX (mIPSC). **r** Averaged mEPSCs for DANs in SNL (*n* = 8, *N* = 1 mice) and SNc (*n* = 11, *N* = 4 mice; *p* = 3.11 × 10⁻⁴, two-sided unpaired Mann-Whitney). **s** Averaged mIPSC for DANs in SNL (*n* = 6, *N* = 2 mice) and SNc (*n* = 14, *N* = 3 mice); *p* = 0.012, two-sided unpaired Mann-Whitney). Box whiskers represent 25–75% percentiles, solid squares represent the mean value, and horizontal box lines represent medians. \**p* < 0.05, \*\**p* < 0.01, \*\*\**p* < 0.001, \*\*\*\**p* < 0.0001. Source data have been uploaded on Zenodo (https://doi.org/10.5281/zenodo.15486331; 2025).

## Functional input mapping reveals exclusive innervation of SNL DA neurons by auditory association cortex

To examine the major inputs to SNL DANs, we first retrogradely labeled projections to the SNL by injecting Cholera Toxin Subunit B conjugated to Alexa Fluor 647 (CTB647) into the SNL (Fig. 2a). We found strong retrograde labeling in multiple brain regions including auditory and visual cortical areas (Supplementary Fig. 11). There was particularly prominent labeling in layers 2/3 and 5 of the auditory

association cortex (AuV/TeA) (Fig. 2a, b and Supplementary Fig. 12), a high-order auditory processing station that integrates auditory infor-mation with experience-dependent cues[40]. Given the strength of labeling in the auditory association cortex, the remainder of our experiments focused on the AuV/TeA neurons.

To functionally examine the impact of these cortical projec-tions, we performed stereotaxic viral injections of AAV-CoChR in AuV/TeA of DAT-Cre Ai9 mice (Fig. 2c) and tested the impact of

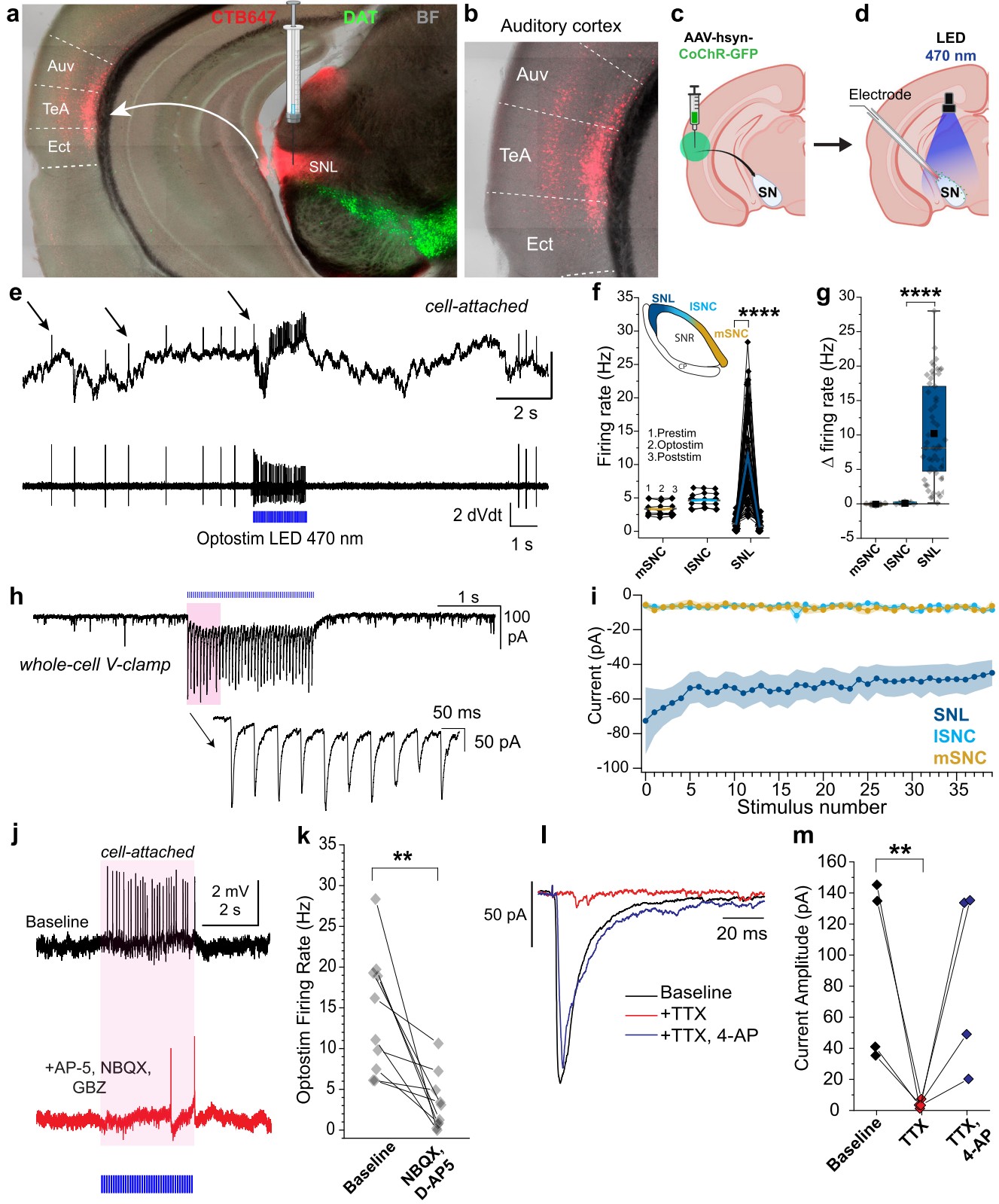

optical stimulation of AuV/TeA projections while performing cell-attached recordings in SNL, lSNc and mSNc DANs (Fig. 2d). Optical stimulation of AuV/TeA inputs (2 ms of 470 nm light pulses delivered at 20 Hz for 2 s) dramatically increased the firing rate of SNL DANs (Fig. 2e) from 1.14 ± 0.11 Hz pre-stimulation to 11.32 ± 0.93 Hz (Fig. 2f; $n = 57$, $p = 2.38 \times 10^{-15}$) corresponding to an average increase of 10.18 Hz ± 0.94 Hz (Fig. 2g). Surprisingly, SNc DANs did not

respond significantly to optical stimulation with mSNc average firing rate going from a baseline of 3.35 ± 0.38 Hz to 3.29 ± 0.37 Hz during optical stimulation and lSNc DANs average firing rate going from a baseline of 4.61 ± 0.44 Hz to 4.70 ± 0.42 Hz during optical stimulation (Fig. 2f; mSNC, $n = 8$, $p = 0.36$; lSNc: $n = 7$, $p = 0.23$) which corresponds to a change in firing rate of −0.05 ± 0.05 Hz and 0.09 ± 0.07 Hz, respectively (Fig. 2g).

**Fig. 2 | Auditory association cortex projects specifically to SNL DA neurons.**
**a** Confocal image of coronal slice from DAT-Cre Ai9 mouse showing DANs (*green*) and retrogradely labeled neurons in the cortex from CTB-647 injection in SNL (*red*). **b** Magnification of auditory association cortex **c, d** Illustrations showing optogenetics and patch-clamp strategy. **e** *Top*, Cell-attached recording from SNL DAN during optical stimulation (470 nm, 2 s, 20 Hz, 2 ms), *arrows* indicating action potentials. *Bottom*, 1st derivative of the trace above showing clear resolution of action potentials. **f** Firing rates before (1), during (2) and after (3) optical stimulation for mSNc, lSNc and SNL DANs. *Black dot*, average firing rate from 5 sweeps from single neurons. *Colored lines*, means. (SNL DANs, prestim vs. optostim, $p = 3.8 \times 10^{-14}$, two-sided paired Sample t-test). **g** Absolute % change (Δ) in firing rate during optical stimulation. Box whiskers represent 25–75% percentiles (SNc vs. SNL Δ firing rate, $p = 2.08 \times 10^{-14}$, two-sided unpaired Mann-Whitney). **h** *Top*, representative voltage clamp recording of SNL DAN responding to optostim with clear oPSC. **i** Averaged oPSCs amplitude for SNL, lSNc and mSNc DANs. Data are

presented as mean ± SEM. **j** Cell-attached recording from SNL DAN during optical stimulation of auditory projections (470 nm, 2 s, 20 Hz, 2 ms) in control (*Top, black*) and in the presence of D-AP5, NBQX, and GBZ (*Bottom, red*) showing firing suppression. **k** Averaged firing rate for SNL DANs during optostim of auditory projections for baseline and in excitatory blockers (Baseline vs. NBQX + D-AP5, $n = 10$, $N = 3$ mice, $p = 0.002$, two-sided paired Wilcoxon Signed Rank Test). **l** SNL DANs oPSC from single light pulse (2 ms, 470 nm) for baseline (solid black), TTX (red) and TTX + 4-AP (blue). **m** Averaged oPSCs amplitude in SNL DANs for baseline, TTX and TTX + 4-AP. (Baseline vs. TTX, $n = 4$, $N = 1$ mice, $p = 0.03$, one-sided paired Sample t-test). Box whiskers represent 25–75% percentiles, solid squares are mean value, horizontal box lines represent medians. *$p < 0.05$, **$p < 0.01$, ***$p < 0.001$, ****$p < 0.0001$. Created in BioRender. Sansalone, L. (2025) https://BioRender.com/7e6etiy, https://BioRender.com/a8o4ql2, https://BioRender.com/ewby07k. Source data have been uploaded on Zenodo (https://doi.org/10.5281/zenodo.15486331; 2025).

We confirmed these findings by testing for the presence of synaptic currents (oPSCs) in response to optical stimulation of the AuV/TeA in the same neurons (Fig. 2h). Importantly, SNL DANs consistently displayed oPSCs while mSNc and lSNc DANs did not show measurable currents upon stimulation (Fig. 2i). When performing voltage-clamp recordings, in the presence of tetrodotoxin (TTX) and 4-aminopyridine (4-AP), SNL DANs consistently responded to a single light pulse with a clear oPSC (Fig. 2l, m; average oPSC amplitude, $n = 4$, 89.11 ± 29.50 pA) demonstrating that projections from AuV/TeA to SNL DANs are monosynaptic in nature.

To further investigate the neurotransmitters and receptors that mediate excitatory responses in SNL DANs upon stimulation of AuV/TeA terminals, we performed cell-attached current-clamp studies to test if blockade of excitatory synaptic receptors was sufficient in preventing the excitation of SNL DANs. Upon optical stimulation of cortical terminals, we found that the firing rate increase is significantly reduced in the presence of D-AP5 and NBQX to block NMDA and AMPA receptors. We obtained consistent responses from 10 different SNL DANs (Fig. 2j-k, avg firing rate in response to optostim; Control, 14.30 ± 2.33 Hz, $n = 10$; D-AP5 + NBQX, 3.26 ± 1.10 Hz, $n = 10$; $p = 0.002$, Paired Wilcoxon Signed-Rank Test). These findings suggest that the excitation of SNL DANs is driven primarily by activation of AMPA and NMDA receptors following glutamate release from cortical terminals.

### DA projections to the tail of the striatum contribute to auditory threat learning
SNL DANs contribute to aversive learning, and their projections to the tail of the striatum (TS) are involved in learning of threat avoidance[12,15,20–22]. Therefore, we next evaluated the role of DA released in TS during auditory threat conditioning[23]. To test this, we inhibited synaptic transmission from DANs projecting to the TS by retrograde viral infection through their terminals with a virus that expresses tetanus toxin light chain (TetTx). DAT-Cre mice were injected in TS with Cre-dependent viruses, either AAV-DJ-DIO-eGFP-TetTx (treated) or AAV-DIO-eGFP (control) and viral expression was confirmed 4 weeks later (Fig. 3a).

Our threat conditioning paradigm (Fig. 3b) consisted of two phases on Day 1 (Context A) - a habituation phase during which pure tones (5 kHz, 30 s duration; conditioned stimulus, CS) were presented alone five times at random intervals, followed by a conditioning phase during which presented pure tones co-terminated with an electric footshock (0.6 mA, 1 s; unconditioned stimulus, US). Twenty-four hours later on Day 2, the conditioned animals were placed in a novel context (Context B, Fig. 3b) and underwent a retrieval phase during which pure tones were presented without footshock to test threat memory expression. For each phase, animal freezing was quantified as the percentage of time that the animal spent freezing during pure tone presentation.

During the habituation phase, we found that both control mice and mice expressing TetTx in TS-projecting DANs showed little freezing in response to pure tones (Fig. 3c) (habituation phase,% freezing; control, 0.19 ± 0.20%, $N = 8$; TetTx, 0.41 ± 0.27%, $N = 8$; $p = 0.73$, Mann-Whitney). However, during the conditioning phase in which tones were paired with footshocks, we observed an overall increase in freezing behavior (Fig. 3c, e). Notably, mice expressing TetTx froze less than controls on average (conditioning phase,% freezing: control, 23.21 ± 5.49%, $N = 8$; TetTx, 7.78 ± 3.46%, $N = 8$). Similarly, during the retrieval phase (Fig. 3d,e), control mice continued to consistently freeze in response to the CS tone while treated mice froze less (retrieval phase,% freezing; control, 47.13 ± 5.63%, $N = 8$; TetTx, 13.32 ± 3.37%, $N = 8$).

Two-way mixed model ANOVA revealed a significant reduction of freezing for treated mice, expressing TetTx in TS-projecting DANs relative to controls across all phases (treatment; $F(1,14) = 14.67$, $p = 0.0018$). Freezing across both groups was different, with higher freezing observed during the retrieval phase (phase $F(1,14) = 13.19$, $p = 0.0027$). Lastly, there was a difference in freezing between TetTx-treated and control mice that was dependent upon phase. Thus, expression of TetTx in TS-projecting DANs resulted in a larger difference in freezing in the retrieval phase relative to the conditioning phase (treatment-phase interaction ($F(1,14) = 6.57$, $p = 0.0225$). Sidak's post hoc comparisons revealed that freezing increased significantly from conditioning to retrieval for the control group (adjusted $p = 0.0013$) but not for the treated group (adjusted $p = 0.7113$). Finally, freezing was significantly different between control and treated groups for both conditioning (adjusted $p = 0.04$; Sidak's post hoc) and retrieval (adjusted $p = 0.0002$; Sidak's post hoc) phases. Therefore, our results demonstrate that inhibition of neurotransmitter release from SNL DANs to TS interferes with the encoding and consolidation of auditory threat memory. These results align with the view that DA in TS plays a role in promoting innate reactions to threatening stimuli.[12,15,20–22]

### Cortical transmission to SNL DA neurons contributes to conditioning and retrieval of auditory threat memories
Our results showed that the auditory association cortex provides strong excitation to SNL DANs; however, the behavioral consequences of cortical input to SNL DANs are unknown. To test the contribution of AuV/TeA->SNL projections in threat learning, we inhibited transmission from SNL projecting cortical neurons in the AuV/TeA by infecting these cells with TetTx. We performed a procedure involving two stereotaxic injections in C57WT mice - the first consisted of retrograde AAV9-Cre[41–44] injected into the SNL to target AuV/TeA neurons, followed by a second injection of AAV-DIO-eGFP-TetTx into the AuV/TeA to selectively express TetTx in SNL-projecting neurons (Fig. 4a, b). Next, we performed Pavlovian threat conditioning over three days

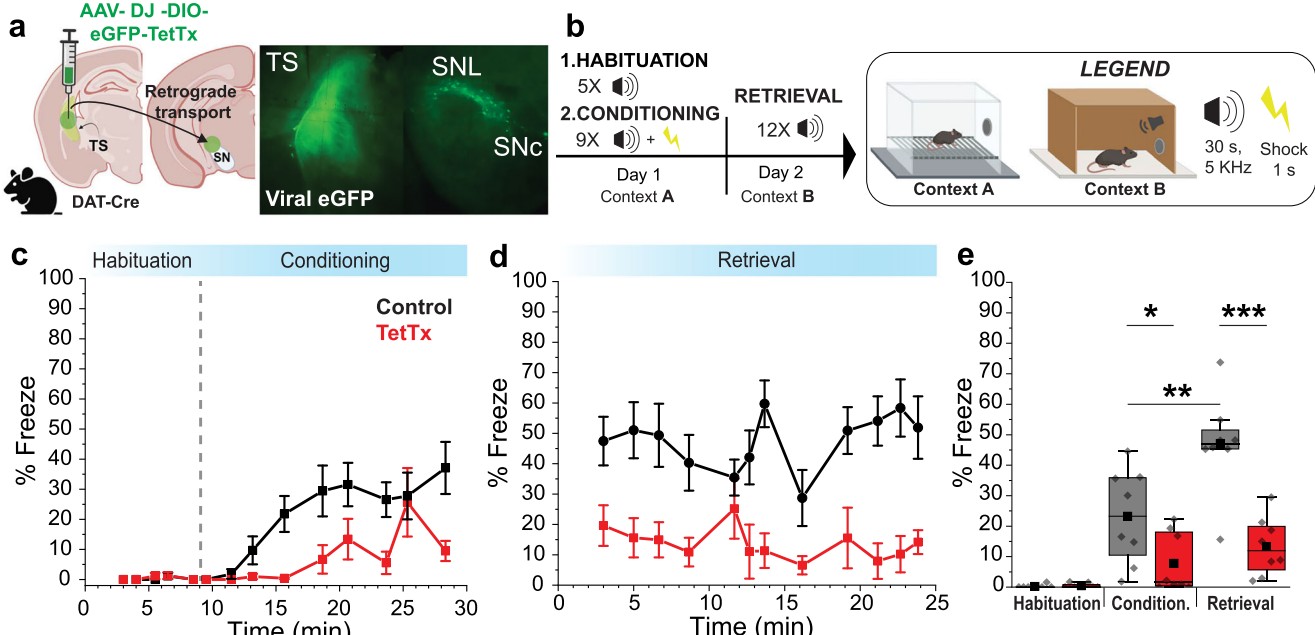

**Fig. 3 | Tail of the striatum (TS) projecting DA neurons contribute to auditory threat learning. a** *Left,* Illustration showing TS injection site for AAV-DIO-TetTx-eGFP or AAV-DIO-eGFP. *Right,* Representative brightfield and fluorescence images showing the extent of viral infection in TS. **b** Behavioral paradigm used for auditory threat conditioning. **c** Graph showing average % freeze during auditory tones for the habituation phase (before dotted line) and conditioning phase (after dotted line). Each squared symbol represents the average for a 30 s pure tone (*N* = 8 for each group). Whisker bars are ± SEM. **d** Graph showing the average % freeze during auditory tones for the retrieval phase. Each squared symbol represents the average for a 30 s pure tone (*N* = 8 for each group). Whisker bars ± SEM. **e** Average % freeze during habituation, conditioning, and retrieval phases for control (eGFP, *N* = 8,

*gray*) and treated (TetTx, *N* = 8, *red*) mice. Two-way ANOVA showed significant treatment (*p* = 0.0018), phase (*p* = 0.0027), and phase-treatment interaction (*p* = 0.0225). Sidak's post-hoc test revealed a significant difference between groups during conditioning (*p* = 0.04) and retrieval (*p* = 0.0002). Box whiskers represent 25–75% percentiles, solid squares are the mean value, and horizontal box lines represent medians. \**p* < 0.05, \*\**p* < 0.01, \*\*\**p* < 0.001, \*\*\*\**p* < 0.0001. DAT^IRESCre (B6.SJL-Slc6a3^tmL1(cre)Bkmn/J) mice were used at 3 months of age, and each experimental group was composed of 50% male and 50% male. Created in BioRender. Sansalone, L. (2025) https://BioRender.com/4cs5d9o, https://BioRender.com/021exyd, https://BioRender.com/d9tvdx3. Source data have been uploaded on Zenodo (https://doi.org/10.5281/zenodo.15486331, 2025).

using the same context (Context A) during habituation, conditioning, and retrieval (Fig. 4c). We used a train of frequency modulated tones (5–20 kHz, 0.5 s tones at 1 Hz for 30 s duration; conditioned stimulus, CS) to assess the function of AuV/TeA->SNL projections because it was shown that this cortical area is important for processing complex (FM) sounds[26,45].

Similar to our earlier experiments, we found that during the habituation phase, both control mice and mice expressing TetTx in SNL-projecting cortical neurons of the AuV/TeA showed little freezing in response to pure tones (Fig. 4d, g) (habituation phase;% freezing, control, 7.04 ± 1.48%, *N* = 8; treated, 3.63 ± 2.05%, *N* = 6; *p* = 0.13, Mann-Whitney). However, we observed diminished freezing for TetTx treated mice during conditioning (Fig. 4e, g;% freeze; conditioning phase - control, 31.61 ± 5.97%, *N* = 8; TetTx treated, 14.73 ± 6.01%, *N* = 6) and retrieval (Fig. 4f, g;% freeze; retrieval phase - control, 31.87 ± 5.54%, *N* = 8; treated, 9.94 ± 2.65%, *N* = 6).

Two-way ANOVA revealed significantly less freezing for treated mice expressing TetTx in SNL-projecting AuV/TeA cortical neurons relative to controls across all phases (treatment; F(1,12) = 8.733, *p* = 0.012). Freezing across both phases was similar (phase; F(1,12) = 0.2882, *p* = 0.6012) with no treatment-phase interaction (F(1,12) = 0.3582, *p* = 0.5607). Indeed, Sidak's post hoc comparisons revealed that freezing did not change significantly from conditioning to retrieval for both control and TetTx-treated groups (control, adjusted *p* = 0.9986; treated, adjusted *p* = 0.7161; Sidak's post hoc). However, freezing was significantly different between control and TetTx-treated groups only during retrieval (adjusted *p* = 0.0193; Sidak's post hoc) but not during conditioning (adjusted *p* = 0.0795; Sidak's post hoc). These results demonstrate that the inhibition of

neurotransmitter release from AuV/TeA terminals to SNL DANs interferes with the association of frequency-modulated auditory tones with threat memory.

The experiments described above were performed using the same context for conditioning and retrieval, which raises the question of whether the freezing during the retrieval phase represents contextual rather than auditory threat conditioning. To distinguish between contextual and auditory threat conditioning, we ran a Pavlovian threat conditioning paradigm with two new animal cohorts with the same treatment (TetTx expression in SNL-projecting AuV/TeA cortical neurons) using different contexts for conditioning (Context A) and retrieval (Context B; Fig. 4h). We used pure tones (5 kHz, 30 s duration; conditioned stimulus, CS) which were presented alone for five times at random intervals during the habituation phase, followed by a conditioning phase where pure tones co-terminated with an electric footshock (0.6 mA, 1 s; unconditioned stimulus, US). The retrieval phase consisted of 10 pure tones presented at random intervals.

During the habituation phase, we found no significant differences in freezing behavior for control and treated mice (Fig. 4i, l) (habituation phase; % freezing, control, 2.96 ± 1.50%, *N* = 8; treated, 0.38 ± 0.22%, *N* = 8; *p* = 0.07, Mann-Whitney). However, we found diminished freezing in TetTx treated mice during conditioning (Fig. 4i, l) (conditioning phase;% freezing, control, 30.50 ± 7.27, *N* = 8; treated, 12.88 ± 3.48%, *N* = 8) and retrieval (Fig. 4j, l) (retrieval phase,% freezing, control, 26.53 ± 10.15%, *N* = 8; treated, 9.18 ± 4.01%, *N* = 8).

Two-way ANOVA revealed significantly less freezing for treated mice expressing TetTx in SNL-projecting AuV/TeA cortical neurons relative to controls across all phases (treatment; F(1,14) = 5.138, *p* = 0.0398). Freezing across both phases was similar (phase;

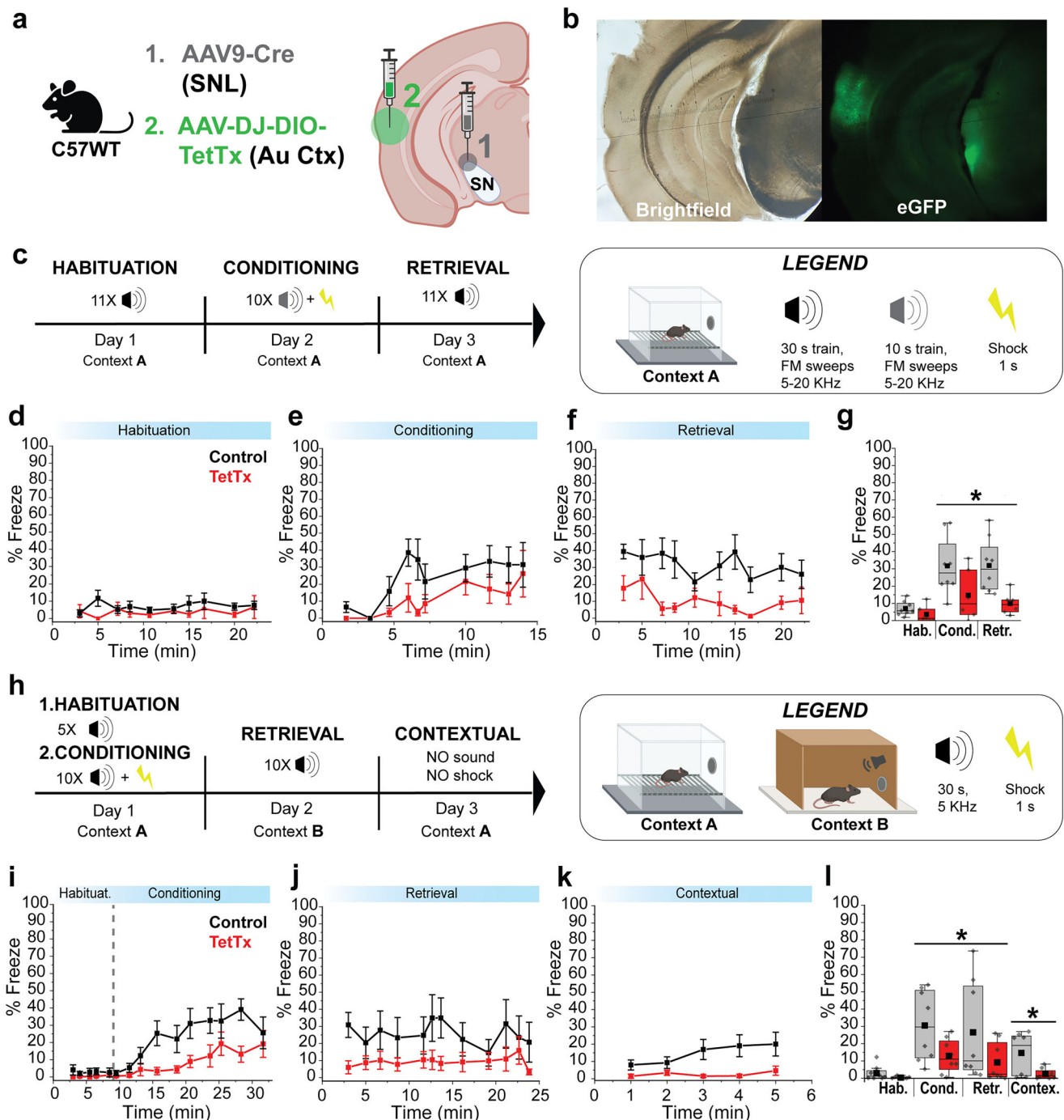

**Fig. 4 | Auditory threat conditioning is modulated by auditory association cortex neurons projecting to SNL. a** Illustration showing infection strategy with 2 viral injections: Retrograde AAV9-Cre in SNL, followed by AAV-DIO-eGFP or AAV-DIO-eGFP-TetTx in AuV/TeA. **b** Representative brightfield and fluorescence images showing the extent of viral infection in AuV/TeA/SNL. **c** Behavioral paradigm used for FM auditory threat conditioning. **d–f** Graphs showing average % freeze for habituation, conditioning, and retrieval phases. Squared symbols represent average % freeze during FM tone trains (control, $N = 8$; TetTx, $N = 6$). Whisker bars are ± SEM. **g** Average % freeze during FM tones for control (eGFP, $N = 8$, *gray*) and treated (TetTx, $N = 6$, *red*) mice. Square symbols represent averages from a single mouse. Two-way ANOVA showed significant treatment ($p = 0.012$), but not phase ($p = 0.60$), or phase-treatment interaction ($p = 0.36$). Sidak's post-hoc test revealed a significant difference between groups during retrieval ($p = 0.02$). **h** Behavioral paradigm used for simple tones auditory threat conditioning. **i–k** Graph showing average % freeze for habituation/conditioning, retrieval, and contextual phases.

Squared symbols represent the freeze average during a single pure tone for habituation/conditioning and retrieval (control, $N = 8$; TetTx, $N = 8$). Whisker bars are ± SEM. **l** Average % freeze during simple tones for control (eGFP, $N = 8$, *gray*) and treated (TetTx, $N = 8$, *red*) mice. Every squared symbol represents the average from a single mouse. Two-way ANOVA showed significant treatment ($p = 0.0398$), but not phase ($p = 0.51$), or phase-treatment interaction ($p = 0.98$). Average % freeze resulted significantly different between control and treated mice for Contextual Freezing ($p = 0.048$, Mann-Whitney). Box whiskers represent 25–75% percentiles, solid squares are mean value, horizontal box lines represent medians. *$p < 0.05$, **$p < 0.01$, ***$p < 0.001$, ****$p < 0.0001$. C57BL/6 J mice were used at 3 months of age, and each experimental group was composed of 50% male and 50% male. Created in BioRender. Sansalone, L. (2025) https://BioRender.com/3uulg98. Source data have been uploaded on Zenodo (https://doi.org/10.5281/zenodo.15486331, 2025).

F(1,14) = 0.4514, *p* = 0.5126) with no treatment-phase interaction (F(1,14) = 0.0005843, *p* = 0.9811). Sidak's post hoc comparisons revealed no change in freezing from conditioning to retrieval for both control and treated mice (control, adjusted *p* = 0.8633; treated, adjusted *p* = 0.8803; Sidak's post hoc). There was also no difference between control and treated mice during conditioning (adjusted *p* = 0.1479; Sidak's post hoc) or retrieval (adjusted *p* = 0.1562; Sidak's post hoc). These results demonstrate that inhibition of neurotransmitter release from AuV/TeA cortical terminals to SNL DANs interferes with auditory threat conditioning.

Finally, we quantified contextual freezing (Fig. 4k) by placing mice in the conditioning cage for five minutes in the absence of sounds or shocks on day 3. We observed a significant difference in freezing behavior between control and treated mice that depends on context (Fig. 4l;% freeze; control, 14.65 ± 4.17%, *N* = 8; treated, 2.54 ± 1.08%, *N* = 8; *p* = 0.048, Mann-Whitney). The disruption of the SNL-projecting TeA cortical neurons may underlie the effect of contextual threat conditioning, which may result from the integrative nature of the TeA[46]. Although the effect of contextual conditioning was significant, the overall effect size was substantially smaller relative to freezing observed in response to auditory tones. Therefore, these experiments demonstrate that inhibition of AuV/TeA to the SNL corticonigral pathway is critical for the associative processing of auditory stimuli with physical threats by encompassing both the learning and establishment phases of Pavlovian threat conditioning.

## Discussion

We demonstrate that dopaminergic neurons (DAN) located in the substantia nigra pars lateralis (SNL) comprise a functionally distinct subpopulation defined by their intrinsic properties, synaptic inputs and behavioral contribution to threat conditioning. Comparison of the intrinsic properties of substantia nigra DANs shows that SNL DANs are distinguished by irregular firing and high maximal firing rates, which likely results from a different ion channel complement compared to SNc DANs. Examination of synaptic projections from different brain nuclei shows that SNL DANs are distinct, receiving different input relative to SNc DANs (Supplementary Fig. 12). We reveal that the auditory association cortex, comprised of secondary auditory cortex (AuV) and temporal association cortex (TeA; Supplementary Fig. 12), provides robust and specific input to SNL DANs, which are likely to contribute to auditory threat conditioning. By contrast, our results show that SNc DANs are not targeted by cortical inputs (Fig. 2f, g, i and Supplementary Fig. 12). Importantly, we show that suppression of synaptic release in auditory association cortex terminals significantly impairs auditory threat conditioning. Thus, our results expand the knowledge on the dopaminergic system in the modulation of animal behavior, demonstrating that aversive responses and threat conditioning likely involve direct communication from cortical projections to DANs in the SNL.

### Intrinsic properties of SNL DANs differ substantially from SNc DANs

Although the firing properties of DAN subpopulations within the substantia nigra pars compacta (SNc) have been well described[47–50], there has been almost no exploration of the intrinsic firing properties of SNL DANs. In this study, we extensively characterized the firing properties of SNL DANs. We found that SNL DANs fire rapidly at substantially higher maximal firing rates relative to canonical SNc DANs, which exhibit low maximal rates as a result of their tendency to enter depolarization block[51]. In addition, we found that spontaneous firing in SNL DANs is irregular, consistent with in vivo extracellular recordings of midbrain DANs[52]. The persistence of irregular firing in SNL DANs, even in the presence of synaptic blockers (Fig. 1o), demonstrates that this is an intrinsic membrane property of these neurons. Moreover, we show that SNL DANs are smaller in

size, have higher maximal firing rate, and display lower levels of HCN and small-conductance calcium-activated potassium (SK) conductances compared to SNc DANs. Importantly, lower expression of SK channels in SNL DANs correlates with irregular firing as observed in ventral tegmental area (VTA) DANs[53–55]. These findings show that SNL DANs are clearly distinguished from other substantia nigra DANs.

The observed differences in the morphology and physiological properties of SNL DANs and those located in the lateral SNc (lSNc) that we report here have implications for understanding the role of dopamine in the tail of the striatum (TS). The intrinsic properties strongly determine the cell's input-output relationship. Thus, our observations raise the possibility that the SNL and lSNc may provide distinct signals to TS, as DANs in both regions project to this area. Moreover, this distinct signaling may also apply to the subset of VGluT2 DANs that we found exhibiting differences between SNL and SNc DANs (Supplementary Fig. 6). Therefore, in future behavioral studies that use VGluT2- or Calb-Cre mouse lines to study dopamine release in TS, it would be interesting to examine separately the contributions of SNL and lateral SNc neurons to threat behaviors.

### Circuit connectivity of SNL DANs is distinguished by robust cortical projections

Our analysis of miniature post-synaptic currents in SNL DANs suggests that they receive predominantly excitatory input, which contrasts with canonical SNc DANs that receive primarily inhibitory synaptic input[39]. This finding supports the idea that DANs in the SNL and SNc participate in different circuits, but also raises the question of which projections may provide unique inputs to SNL DANs. Our study revealed robust corticonigral projections from the auditory association cortex to SNL DANs, making these neurons unique among midbrain DANs. Following the earliest proposals of a corticonigral pathway from a century ago, there have been many studies that have tested for its existence, but the results were unclear. The most compelling evidence has been provided by anatomical studies that report only sparse connections from cortex to SNc DANs[27,56]. Functional evidence for corticonigral projections has been very limited and often ambiguous, particularly in in vivo experiments, due to possible indirect excitation through STN. A recent study tested motor cortex (M1 and M2) inputs to SNc DANs and found that optogenetic activation of M1/M2 projections resulted in weak responses in a small number of SNc DANs[30]. By contrast, our findings here demonstrate that the vast majority of SNL DANs exhibit strong increases in firing upon optical stimulation of auditory association cortex projections compared to SNc DANs, which did not show any functional responses. We demonstrate that SNL DANs represent the only substantia nigra region receiving strong monosynaptic input from auditory association cortex (Fig. 2 and Supplementary Fig. 12). Thus, corticonigral input likely contributes to the higher frequency of mEPSCs that were observed in SNL compared to SNc DANs.

To further determine if other major cortical areas project to SNL, we characterized retrograde labeling from CTB 647 injections in the SNL (Supplementary Fig. 11). In addition to auditory association areas (secondary auditory and temporal association cortices; AuV/TeA), we found labeling in visual cortex (Supplementary Fig. 11b) and very sparse labeling in insular cortex (Supplementary Fig. 11d). Surprisingly, we did not find any labeling in somatosensory and motor areas such as S1 or M1 (Supplementary Fig. 11c, d). While these findings support our auditory threat conditioning results, labeling in the visual cortex supports recent findings that implicate SNL in looming-based behavior[57]. Whether visual cortex projections form functional synapses onto SNL DANs and what the function of these inputs is in the context of learning behaviors remains unknown and represents an interesting question for future studies.

## Corticonigral projections recruit SNL DA neurons during auditory threat conditioning

Our data show that tetanus toxin inhibition of synaptic transmission from AuV/TeA to SNL significantly reduces animal freezing during Pavlovian threat conditioning paradigms, suggesting that this corticonigral pathway is important for auditory threat memory encoding and consolidation. This result is supported by previous work showing that cortical neurons located in AuV/TeA are critical for memory acquisition and expression[26,58]. Interestingly, it has been shown that the primary auditory cortex responds to frequency modulated (FM) sweeps only, while AuV/TeA responds to both FM sweeps and pure tones[26], thus supporting our results (Fig. 4).

Past work has identified AuV/TeA neurons that are responsive to frequency-modulated sweeps and project to the lateral amygdala, while other AuV/TeA neurons are involved in pure tone conditioning and likely use different output pathways. Our work here reveals that cortical neurons in AuV/TeA send functional projections to SNL DAN. This result, together with our behavioral findings that auditory threat conditioning is dependent on SNL DANs and can be driven both by FM sweeps and pure tones, demonstrates that both previously identified populations of AuV/TeA neurons synapse onto SNL DANs. Finally, our data is further supported by previous work showing that cortical neurons located in layers 2, 3 and 5 of AuV/TeA exhibited activity-dependent changes in excitatory genes in mice that underwent threat conditioning paradigms[58].

## Implications for motor and psychiatric disorders

Our study uncovers the physiological properties and circuit connectivity of SNL DANs, which are not directly involved in motor control, unlike SNc DANs. During PD neurodegeneration, SNc DANs are lost while SNL DANs are relatively spared[59]. Here, we demonstrate clear differences in the intrinsic properties and circuits that surround these distinct subpopulations, which may provide insight into the selective vulnerability of dopamine neurons to cell death in PD. In addition, most PD studies focus on motor symptoms, but there is less understanding as to why non-motor symptoms appear in PD patients. Our work uncovers a previously unknown function of dopamine-producing neurons in the SNL and can help to understand the pathophysiology underlying non-motor symptoms in PD patients.

Importantly, our data may have implications for understanding human psychiatric disorders such as post-traumatic stress disorder (PTSD) and phobias, where misevaluation of the threat value of sensory stimuli may trigger abnormal physiological reactions. The severity of clinical symptoms in PTSD and phobia has been linked to sensory areas[31], suggesting that sensory processing dysfunction might contribute to symptoms' severity. Importantly, the unique properties of SNL DANs, including faster firing and higher input resistance, may contribute to rapid sensory responses that may be advantageous for threat learning. Here, we report that the interaction of auditory association cortex and dopaminergic neurons in the SNL contributes to threat conditioning, raising the possibility that this pathway may be involved in circuit dysfunction in PTSD and phobia. Altogether, our findings provide a framework that may help future studies to identify the mechanisms underlying cortical sensory processing in downstream areas during aversive behaviors.

## Methods

### Ethical compliance

All procedures were conducted in accordance with the guidelines established by the Animal Care and Use Committee for the National Institute of Neurological Disorders and Stroke and the National Institutes of Health. All mice were housed and bred in a vivarium with standard laboratory chow and water in cages holding 1-5 animals. Light followed a 12 h circadian cycle.

### Animals

Experiments were carried out using male and female mice at 4–24 weeks of age. The following strains were used: DAT[IRESCre] (B6.SJL-Slc6a3[tm1.1(cre)Bkmn]/J), The Jackson Laboratory, Strain #: 006660; VGluT2[IRESCre] (Slc17a6[tm2(cre)Lowl]/J), The Jackson Laboratory, Strain #: 016963; Calb1-[IRES2-Cre-D] (B6;129S-Calb1[tm2.1(cre)Hze]/J), The Jackson Laboratory, Strain #: 028532); DAT-Flp (Slc6a3[em1(flpo)Hbat]/J)[60], The Jackson Laboratory, Strain #: 035436; Ai65(RCFL-tdT)-D (B6.129S-Gt(ROSA)26Sor[tm65(CAG-tdTomato)Hze]/J, The Jackson Laboratory, Strain #: 021875); Ai9(RCL-tdT) (B6.129S6-Gt(ROSA)26Sor[tm9(CAG-tdTomato)Hze]/J, The Jackson Laboratory, Strain #: 007905; C57BL/6 (C57BL/6NCrl, Charles River Laboratories, Strain #: 027); Tyrosine hydroxylase-GFP (Th-GFP; C57BL/6 background[38,60]).

### Stereotaxic surgery

All stereotaxic injections were conducted on a Stoelting QSI (Cat# 53311). Mice were maintained under anesthesia for the duration of the injection with 1.5% isoflurane and allowed to recover from anesthesia on a warmed pad. At the end of the injection, the needle was raised 1–2 mm for a 10-min duration before the needle was removed. All mice were allowed to recover in their cage after injections and received subcutaneous ketoprofen (10 mg/kg) for 3 consecutive days post-injection. Mice were used for ex vivo electrophysiology or behavioral experiments 3–5 weeks after injections. The retrograde tracer cholera-toxin subunit B conjugated to Alexa fluor™ 647 (CTB-647, ThermoFisher Scientific Cat # C34778) was bilaterally injected into the SNL (ML: ± 2.0, AP: − 3.0, DV: − 4.1) of C57WT or DAT-Cre Ai9 mice. AAV-CoChR (AAV1-hSyn-CoChR-GFP, UNC vector core, Boyden) was injected bilaterally into AuV/TeA (ML: ± 4.5, AP: − 3.0, DV: − 3.4) of DAT-Cre Ai9 mice.

AAV-DIO-eGFP-Tet-LC (AAVDJ-CMV-DIO-eGFP-2A-TeNT, Stanford University Gene Vector and Virus Core, Cat #GVVC-AAV-71) and AAV-DIO-eGFP (AAVDJ-CMV-DIO-eGFP, Stanford University Gene Vector and Virus Core, Cat #GVVC-AAV-12) were injected bilaterally into TS (ML: ± 3.4, AP: − 0.7, DV: − 3.0) and AuV/TeA (ML: ± 4.5, AP: − 3.0, DV: − 3.4) of C57WT mice. AAV9-Cre (AAV0-hSyn-Cre-WPRE-hGH, Addgene Cat # 105553) was injected bilaterally into the SNL (ML: ± 2.0, AP: − 3.0, DV: − 4.1) of C57WT mice.

### Slice preparation

Mice were anesthetized with isoflurane, decapitated, and their brains extracted. Coronal midbrain slices (200 μm) were prepared using a vibratome (Leica VT1200S). Slices were cut in ice-cold, oxygenated, slicing solution containing the following (in millimolar (mM)): 198 glycerol, 2.5 KCl, 1.2 NaH2PO4, 20 HEPES, 25 NaHCO3, 10 glucose, 10 MgCl2, 0.5 CaCl2, 5 Na-ascorbate, 3 Na-pyruvate, and 2 thiourea. Slices were then incubated for 30 min at 34 °C in oxygenated holding solution containing the following (in millimolar (mM)): 92 NaCl, 30 NaHCO3, 1.2 NaH2PO4, 2.5 KCl, 35 glucose, 20 HEPES, 2 MgCl2, 2 CaCl2, 5 Na-ascorbate, 3 Na-pyruvate, and 2 thiourea. Slices were then stored in the same holding solution at 20–25 °C, with constant carbogen perfusion, and electrophysiological recordings were performed within 1 h to 6 h.

### Electrophysiological recordings

Slices were continuously superfused at 2.7 ml/min with warm (34 °C), oxygenated extracellular aCSF recording solution containing the following (in millimolar (mM)): 125 NaCl, 25 NaHCO3, 1.25 NaH2PO4, 3.5 KCl, 10 glucose, 1 MgCl2, and 2 CaCl2 (Osmolarity: 290–310 mOsm). Neurons were visualized with a 60x objective using a BX61WI Olympus microscope equipped with a Hamamatsu digital camera ORCA-ER (C4742-80). Recordings were obtained using low- resistance pipettes (2.2 − 5 MΩ) pulled from filamented borosilicate glass (World Precision Instruments, Cat #1B150F-4) with a flaming/brown micropipette puller (Sutter Instruments, Model P-97). Cell-attached and whole-cell current-clamp recordings were made using

borosilicate pipettes filled with internal solution containing (in mM) 122 KMeSO3, 9 NaCl, 1.8 MgCl2, 4 Mg-ATP, 0.3 Na-GTP, 14 phosphocreatine, 9 HEPES, 0.45 EGTA, 0.09 CaCl2 (Osmolarity: 280 mOsm). Some experiments included 0.1% − 0.3% neurobiotin (Vector Laboratories, Inc., Cat # SP-1120) in the internal solution for post hoc visualization. Whole-cell voltage-clamp recordings were made using borosilicate pipettes filled with internal solution containing (in mM) 120 CsMeSO3, 20 Tetraethylammonium chloride, 2 MgCl2, 4 Mg-ATP, 0.3 Na-GTP, 14 phosphocreatine, 10 HEPES, 10 EGTA, 2 QX314, 0.03 ZD7288 (Osmolarity: 280 mOsm) and cells were held at − 70 mV for AMPA/GABA currents and +40 mV for NMDA currents. For hyperpolarization-activated current (Ih) recordings, voltage-clamp experiments were made using borosilicate pipettes filled with internal solution containing (in mM) 130 KMeSO3, 30 Tetraethylammonium chloride, 10 NaCl, 2 MgCl2, 4 Mg-ATP, 0.3 Na-GTP, 14 phosphocreatine, 10 HEPES, 10 BAPTA-K (Osmolarity: 280 mOsm). Access resistance was monitored, and recordings with $R_a > 25 M\Omega$ were discarded. Liquid junction potential (− 8 mV) was not corrected. All experiments were conducted between 33–36 °C.

### Optogenetics experiments
Whole-field optogenetic activation of CoChR-infected axons in the brain slice was achieved by either a white LED (Prizmatix) sent through a FITC filter (HQ-FITC; U-N41001; C27045) or a blue (470 nm) LED (Thorlabs, LED4D067) sent to the tissue via a silver mirror or through the FITC filter. Light intensity measured at the objective back aperture ranged from 1 − 25 mW. Light activation was given as a single pulse lasting 2 ms or as a 2 s, 20 Hz train with 2 ms pulses, unless otherwise specified.

### Reagents
Patch-clamp recordings and optogenetics experiments, where indicated, were performed in the presence of one or more of the following drugs: 20 mM 2,3-Dioxo-6-nitro-7-sulfamoyl- benzo[f]quinoxaline (NBQX) or 6-Cyano-7-nitroquinoxaline-2,3-dione (CNQX) to block AMPA receptors, 50 mM (2 R)-Amino-5-phosphonopentanoate (D-AP-5) to block NMDA receptors, 50 mM Picrotoxin (PTX) or 10 μM Gabazine (GBZ) to block GABA_A receptor, 1 mM Tetrodotoxin (TTX) to block voltage-gated sodium channels, 200 μM 4-Aminopyridine (4-AP) to block voltage- gated potassium channels, 500 nM CGP-55845 to block GABA_B receptors, 200 nM Apamine to block SK channels. Glucose, glycerol, and salts used to make slicing, holding and perfusing aCSF solutions were purchased from Sigma-Aldrich. Drugs were purchased from Tocris Bioscience and Sigma-Aldrich. All drugs were reconstituted as indicated by the manufacturer and prepared as aliquots in deionized water or DMSO and stored at −20 °C or at −80 °C.

### Data analysis, statistics and reproducibility
Signals were digitized with a Digidata 1590 interface, amplified by a Multiclamp 700B amplifier, and acquired using pClamp 13 software (Molecular Devices). Data were sampled at 50 kHz and filtered at 10 kHz (or 2 kHz for analysis of mEPSC frequency). Data were analyzed using custom code procedures in IgorPro 9 (WaveMetrics), Easy Electrophysiology 2.4, Clampex 11.2 or GraphPad Prism 9. Unless otherwise specified, Mann-Whitney U tests (unpaired) or t-tests were used to compare two groups. For repeated comparisons, paired t-test or the Wilcoxon tests determined significance of the dataset. Data in text is reported as Mean ± SEM, and error bars are ± SEM unless otherwise specified. Boxplots show medians, 25th and 75th (boxes) and outliers 1.5 IQR (whiskers) percentiles. For electrophysiological recordings, biological replicates include neuron samples from at least 3 separate mice unless otherwise specified. For behavior, biological replicates include data from at least 6 mice. Threshold conditioning acquisition and analysis were performed with FreezeFrame 5. Exact P values and sample sizes are provided in the text or figure legends. No statistical method was used to predetermine sample size. No data were excluded from the analyses.

The experiments were not randomized. The Investigators were not blinded to allocation during experiments and outcome assessment. Sex was not considered in the study design; however, initial cohorts of mice used for behavioral experiments included approximately equal numbers of males and females. In electrophysiological experiments, both males and females were used for all experiments. All recordings were performed in DANs. DANs were targeted by their anatomic location, size, and presence of the fluorescence reporter tdTomato in genetically-modified mouse lines (DAT-Cre, DAT-Flp Calb-Cre Ai65, DAT-Flp VGlut2-Cre Ai65). Some elements in the following figures were created with BioRender and licenses have been obtained: Fig. 2a and Supplementary Fig. 11a (https://BioRender.com/7e6etiy), Fig. 2b (https://BioRender.com/a8o4ql2), Fig. 2c (https://BioRender.com/ewby07k), Fig. 3a (https://BioRender.com/4cs5d9o), Fig. 2c (https://BioRender.com/4cs5d9o), Figs. 3b, 4c, 4h (https://BioRender.com/021exyd and https://BioRender.com/d9tvdx3), Fig. 4a (https://BioRender.com/3uulg98), Supplementary Fig. 3 (https://BioRender.com/lnw5xiq).

### Fluorescence in situ hybridization (FISH)
In situ hybridization was performed on 16 μm thick midbrain slices from a fresh-frozen mouse brain cut on a cryostat (Leica CM3050 S). All FISH reagents used are commercially available from ACD Bio, and procedures for the Multi-Plex FISH process were followed as recommended on ACDbio.com. Channels used for this study were TH, DAT (Slc6a3), Calb1 and VGluT2 (Slc17a6). Slices were imaged on a Zeiss LSM 800 using Zen Blue software.

### Auditory threat conditioning and analysis
Pavlovian auditory threat conditioning in Fig. 3 consisted of one continuous habituation- conditioning session (Session 1) occurring in Context A, followed by one retrieval (or recall) session, 24 h later, occurring in Context B (Session 2). Context A was a cage equipped with a speaker and floor-placed parallel metal rods that could deliver foot shocks. Context B was a cage equipped with a speaker and a plastic panel (white) on top of the metal rods, as well as black and white design panels on the side walls. Context B was cleaned with acetic acid before the mice placement. Between sessions, each chamber was cleaned with a 70% ethanol solution. During session 1, 5 tones (CS, 5 kHz, 30 s, 75-80 dB) were delivered randomly with intertone intervals of 60 − 180 s, after which 9 tones (5 kHz, 30 s, 75-80 dB) co-terminating with a foot shock (US, 0.6 mA, 1 s) were randomly delivered. During session 2, 12 tones (5 kHz, 30 s, 75−80 dB) were randomly delivered with intertone intervals of 60 − 180 s. Experiments and data analysis were carried out using Freezeframe 5 (Actimetrics). The percentage of time spent freezing (% freezing) was calculated using Freezeframe software and represents the average time the animal freezes during the tones (CS) duration. Freezing thresholds were manually determined based on software recommendations to prevent quantification of simple resting-pause positions instead of real freezing behavior. Pavlovian auditory threat conditioning in Fig. 4h–l maintained the same protocol as Fig. 3. For Fig. 4c–g, the behavioral paradigm consisted of habituation, conditioning and retrieval sessions performed separately on 3 consecutive days and in the same context (Context A). For habituation and retrieval, 10 frequency modulated (FM) sound trains made of 0.5 s tones were delivered at 1 Hz (5–20 KHz, 30 s, 75–80 dB) with random intertrain intervals of 60 − 180 s. For conditioning, 10 FM sound trains made of 0.5 s tones were delivered at 1 Hz (5–20 KHz, 30 s, 75–80 dB) with random intertrain intervals of 60 − 180 s.

### Reporting summary
Further information on research design is available in the Nature Portfolio Reporting Summary linked to this article.

## Data availability
Source data for figures were deposited on Zenodo (Lorenzo Sansalone, Zayd Khaliq; Corticonigral projections recruit substantia nigra pars

lateralis dopaminergic neurons for auditory threat memories; Zenodo, https://doi.org/10.5281/zenodo.15486331, 2025).

## Code availability
Codes used for data analysis in this study are available from the first author and/or corresponding author upon request.

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

## Acknowledgments

We thank the members of the Khaliq laboratory and Dr. Heather Cameron (NIMH) for their insightful discussions and comments on this manuscript. We thank Dr. Helen Bateup for providing the DAT-Flp mouse line. Funding for this work was provided by Aligning Science Across Parkinson's (ASAP-020529) to Z.M.K. through the Michael J. Fox Foundation for Parkinson's Research (MJFF). This research was also supported by the Intramural Research Program of the National Institutes of Health (NIH). This included a fellowship to L.S. from the Center for Compulsive Behaviors (CCB) as well as funding to Z.M.K through the National Institute of Neurological Disorders and Stroke (NS003135). The contributions of the NIH author(s) were made as part of their official duties as NIH federal employees, are in compliance with agency policy requirements, and are considered Works of the United States Government. However, the findings and conclusions presented in this paper are those of the author(s) and do not necessarily reflect the views of the NIH or the U.S. Department of Health and Human Services.

## Author contributions

L.S. and Z.M.K. conceived the project, designed experiments, and interpreted the data. L.S., E.T., R.C.E., and Z.M.K. analyzed the data. L.S. wrote the first draft of the manuscript. L.S. and Z.M.K. edited and wrote the manuscript. L.S., E.T., R.C.E., and A.B.-G. performed experiments. R.Z. and L.S. performed stereotaxic surgeries.

## Funding

## Competing interests

The authors declare no competing interests.
