## [Transparent Peer Review file · Nature Communications]

Corticonigral projections recruit substantia nigra pars lateralis dopaminergic neurons for auditory threat memories

Corresponding Author: Dr Zayd Khaliq

Version 0:

Reviewer comments:

Reviewer #1

(Remarks to the Author)

Sansalone and colleagues investigate circuit and functional questions related to dopamine neurons in the pars lateralis (SNL) compartment of the substantia nigra. Overall this population has received relatively little direct attention, and the existing data indicate they form unique class of dopamine neurons. These studies add a lot of new understanding to that notion. First, the authors show that sensory cortical areas – here in the temporal/auditory area of mouse cortex, project preferentially to the lateral dopamine neurons, avoiding the more classic compacta located ones. Further, suppression of activity of the lateral dopamine neurons, or of cortical inputs, disrupted different aspects of threat learning.

These studies offer a nice integration of different physiological, circuit, and behavioral techniques and give important insight into an understudied system – the emerging lateral dopamine-tail of the striatum literature offers new ways of thinking about dopamine. I have a few comments that might help to clarify some remaining questions in revision.

The specificity of the cortical inputs to the SNL is interesting. It is unclear from the current paper if other cortical areas showed retrograde labeling to either the SNL or SNC. A little more discussion of this, and perhaps more histology examples/analysis in a deeper supplemental figure showing (I would assume but am not certain), a lack of cortical labeling in other major sensory/association areas would be helpful for understanding the broader significance of this pathway. If the “top-down” influence of SNL is specific to auditory-related cortex, that has interesting implications for the overall function of this system, relative to the SNC (and VTA). For example – one wonders if the behavioral effects reported here specific to auditory (vs visual, odor, etc) threat conditioning? I would not suggest doing the additional experiments to determine that for this paper, but more context for this notion in revision would be clarifying.

The inclusion of the STN and PPN input experiments are certainly interesting and useful for fleshing out the unique circuit regulation of the SNL vs SNC dopamine neurons, but because those inputs are not functionally tested in the subsequent behavioral tasks, these data feel disconnected from the main thrust of the paper. To be clear, I do not suggest that behavioral experiments for those inputs are needed for this paper, but perhaps this section of the data could be better incorporated into the broader narrative. Given the paper title, my take is that the authors thing the most interesting framing relates to cortical regulation of a unique dopamine population.

I am confused by the viral approach used for the cortical-SNL behavior experiment. The paper says AAV9-cre was injected into the SNL and a cre-dependent vector into the auditory cortex. I am not aware of AAV9 being used as a primarily retrograde-traveling serotype – can the authors clarify this?

Reviewer #2

(Remarks to the Author)

This manuscript investigated the intrinsic properties, synaptic inputs, and threat conditioning behavioral contributions of substantia nigra pars lateralis dopaminergic neurons (SNL DANs). This study is significant because in contrast to SN pars

compacta (SNC), the function of SNL DANs is less well understood. In particular, the intrinsic properties and excitatory inputs that drive SNL DAN firing are poorly characterized. The authors identified SNL DANs as a functionally distinct subpopulation characterized by irregular pacemaking and significantly higher maximal firing rates compared to SNC DANs. A key finding was that, unlike SNC DANs, SNL DANs receive robust inputs from auditory association cortices. In behavioral experiments, the authors found that disrupting synaptic release from the auditory cortex to SNL DANs impaired the retrieval of auditory threat memories, but interpreted their data as showing that threat conditioning was unaffected. For the most part this is a well-structured and well-written study that makes a novel and important contribution to the dopamine research field by elucidating the function of an under-studied dopaminergic subpopulation.

However, there is one large weakness of the study in its present form: the noisy nature of the data in Fig. 5 does not adequately support the claim that “preventing synaptic release from auditory cortex to SNL DANs did not affect threat learning”. In fact, the authors themselves acknowledged a “clear trend towards diminished freezing behavior during conditioning” (page 7). Looking at the data in Fig. 5e, the difference seems more than a mere trend (one time point even has a star above it, which suggests statistical significance). Clarifying this issue would be important for strengthening the rigor of the study (it would have no effect on the study’s significance). This could be resolved through additional experiments to increase the sample size. If this is not possible, then the authors would need to tone down the claim that threat learning was not affected. At best, the present data suggests that threat learning is affected by a smaller amount than the retrieval phase.

A moderate weakness was the lack of sufficient information regarding the statistical comparisons used to analyze the results. The p-values and type of test used should be included in the figure captions.

Minor comment:

Typo on page 7, should be “we recorded a significant increase in % freezing...”

Reviewer #3

(Remarks to the Author)

The authors investigated the role of substantia nigra pars lateralis (SNL) dopaminergic neurons (DANs) in auditory threat conditioning. Using functional mapping, optogenetics, and slice electrophysiology, they characterized the firing properties of SNL DANs, identified excitatory inputs from the auditory association cortex to SNL DANs, and examined the functional roles of their projections to the tail of the striatum (TS). They reported that disrupting synaptic transmission in these projections impaired auditory threat learning and memory retrieval, suggesting the importance of the SNL-TS pathway in aversive learning.

To meet the standards for publication in Nature Communications, several key issues need to be addressed. Below, I outline major concerns that should be resolved to strengthen the manuscript.

1. Identification of Neurotransmitters Released by the Auditory Cortex onto SNL DANs (Figure 2):

The study highlights the specific projection of the auditory cortex to SNL DANs. However, the neurotransmitters released by the auditory cortex onto DANs and the mechanisms underlying this communication remain unclear. Further investigation into the specific neurotransmitters and receptor types involved would greatly enhance our understanding of this circuitry.

2. Introduction of STN and PPN (Figure 3):

The authors introduce the subthalamic nucleus (STN) and pedunculopontine nucleus (PPN) as distinct inputs to SNC and SNL, but their functional role in auditory threat learning has not been investigated. This inclusion feels abrupt and underdeveloped. Further examination of their role would enhance the coherence of the study. Additionally, the schematic in Figure 6 should integrate the functional roles of STN and PPN to provide a more complete picture.

3. TetTx Injection (Figure 4):

A critical issue lies in the authors’ approach to silencing synaptic transmission from SNL to the TS. TetTx was injected into the TS using DAT-Cre mice, which silenced DAT-expressing neurons in the TS, but not in the SNL, because AAV entered into the cell bodies and expressed TetTx in the infused area. To effectively address whether disrupting the SNL-TS pathway interferes with auditory threat learning, TetTx should have been injected into the SNL via a projection-specific manner. The current results from this experiment are therefore inconclusive, and further experimentation is needed to support these claims.

4. Threat Conditioning Protocol (Figure 5):

While TetTx was used correctly in this figure, the threat conditioning protocol has inconsistencies. The tone durations during habituation (30 s) and conditioning (10 s train) differ significantly, which may introduce confounds. Additionally, the protocol would have been more robust if the authors had used a Context ABA design. Furthermore, the schematic in Figure 5i does not accurately reflect the described protocol and should be revised for clarity.

5. Minor comments:

- Any abbreviations (e.g., HCN) should be introduced where they first appear, not later in the discussion.
- The terms “DAN” and “DA neurons” are used interchangeably. It would be clearer to choose one term and use it consistently throughout the manuscript.

Reviewer #4

(Remarks to the Author)

Version 1:

Reviewer comments:

Reviewer #1

(Remarks to the Author)

Thanks to the authors for their time in revising this paper. Overall the paper has been heavily revised and updates, including multiple new experiments and analyses. I think the paper is overall stronger and the previous comments have been addressed.

Reviewer #2

(Remarks to the Author)

I am satisfied with the changes the authors made to address my concern about former Fig. 5. The new results and interpretation enhance the rigor and significance of the study. I have no further concerns. The study presents interesting and novel findings on the synaptic properties and functional role of a little-studied population of dopaminergic neurons. One minor suggestion is to improve the wording of the title in Fig. 3, which is currently a bit vague.

Reviewer #3

(Remarks to the Author)

All my concerns have been clearly resolved now. Thank you to the authors for the thoughtful and thorough revisions. I am happy to recommend the manuscript for acceptance.

Reviewer #4

(Remarks to the Author)

We thank the reviewers for the thoughtful and constructive comments. In response to the reviewer's comments:

- 1) We performed new behavioral experiments that more robustly tested the contribution of AuV/TeA cortical projections to the SNL in threat conditioning.
- 2) We also performed additional electrophysiological experiments to identify the receptors that mediate corticonigral excitation.
- 3) We performed anatomical experiments to determine all cortical regions that project to the SNL using retrograde labeling.

In addition, we have revised the manuscript based on these new findings. We believe that the revised manuscript has been significantly improved by the new experiments and we believe that we have comprehensively addressed the concerns of the reviewers.

Below, we have provided a point-by-point response to the reviewer's concerns:

Reviewer #1 (Remarks to the Author):

Sansalone and colleagues investigate circuit and functional questions related to dopamine neurons in the pars lateralis (SNL) compartment of the substantia nigra. Overall this population has received relatively little direct attention, and the existing data indicate they form unique class of dopamine neurons. These studies add a lot of new understanding to that notion. First, the authors show that sensory cortical areas – here in the temporal/auditory area of mouse cortex, project preferentially to the lateral dopamine neurons, avoiding the more classic compacta located ones. Further, suppression of activity of the lateral dopamine neurons, or of cortical inputs, disrupted different aspects of threat learning.

These studies offer a nice integration of different physiological, circuit, and behavioral techniques and give important insight into an understudied system – the emerging lateral dopamine-tail of the striatum literature offers new ways of thinking about dopamine. I have a few comments that might help to clarify some remaining questions in revision.

The specificity of the cortical inputs to the SNL is interesting. It is unclear from the current paper if other cortical areas showed retrograde labeling to either the SNL or SNC. A little more discussion of this, and perhaps more histology examples/analysis in a deeper supplemental figure showing (I would assume but am not certain), a lack of cortical labeling in other major sensory/association areas would be helpful for understanding the broader significance of this pathway. If the “top-down” influence of SNL is specific to auditory-related cortex, that has interesting implications for the overall function of this system, relative to the SNC (and VTA). For example – one wonders if the behavioral effects reported here specific to auditory (vs visual, odor, etc) threat conditioning? I would not suggest doing the additional experiments to determine that for this paper, but more context for this notion in revision would be clarifying.

Thank you for this suggestion. **We performed new anatomical experiments to comprehensively identify cortical regions that project to SNL using CTb retrograde labeling.** Consistent with our previous conclusion, we found the strongest

labeling in auditory association cortex. In addition, we found moderate labeling in the visual and insular cortices. By contrast, we found no labeling in the motor and somatosensory cortex (M1/S1). Thus, the input to the SNL is dominated by projections from auditory association cortex but also involves input from multiple cortical areas including visual cortex. This data is provided in Extended Data Fig. 11.

Regarding the question of whether the behavioral effects are specific to auditory cortex, we believe based on the anatomy that the effects are primarily mediated by auditory cortex. However, we also observed in our new behavioral experiments that TetTx treated mice displayed less freezing in a contextual threat association task (Figure 4k), which may be consistent with the associative nature of the AuV/TeA cortical region as previously reported (Zingg et al. 2014). We have included a section in the revised manuscript discussing the integrative nature of the AuV/TeA (Lines 305-314).

Extended data figure 11 I a, Image showing coronal section from a mouse that was injected in SNL with CTB647 (red). **b-e**, Images showing progressive coronal sections with retrograde labeling in different brain areas. **b-d2**, Images showing coronal sections with labeled neurons that send axonal projections to SNL. AID = agranular insular cortex dorsal, AIV = agranular insular cortex ventral, Au1 = primary auditory cortex, AuV = secondary auditory cortex ventral, DI = dysgranular insular cortex, EcT = ectorhinal cortex, Gl = granular insular cortex, Ins = insular cortex, M1 = primary motor cortex, RSD = retrosplenial dysgranular cortex, S1 = primary somatosensory cortex, S2 = secondary somatosensory cortex, TeA = temporal association cortex, V1 = primary visual cortex, V2L = secondary visual cortex lateral area.

The inclusion of the STN and PPN input experiments are certainly interesting and useful for fleshing out the unique circuit regulation of the SNL vs SNC dopamine neurons, but because those inputs are not functionally tested in the subsequent behavioral tasks, these data feel disconnected from the main thrust of the paper. To be clear, I do not suggest that behavioral experiments for those inputs are needed for this paper, but perhaps this section of the data could be better incorporated into the broader narrative. Given the paper title, my take is that the authors think the most interesting framing relates to cortical regulation of a unique dopamine population.

We agree that the experiments examining the inputs from STN and PPN are disconnected from the main thrust of the paper (similarly commented by Reviewer 3). We believe that these experiments would be better integrated into a separate manuscript that focuses primarily on the contribution of STN and PPN in control of SN DANs in circuit function and behavior. Thus, we have removed all data and discussion related to STN and PPN projections from the revised manuscript.

I am confused by the viral approach used for the cortical-SNL behavior experiment. The paper says AAV9-cre was injected into the SNL and a cre-dependent vector into the auditory cortex. I am not aware of AAV9 being used as a primarily retrograde-traveling serotype – can the authors clarify this?

We agree that this should have been more clearly described. Past work has established that AAV9 serotype can infect both locally and retrogradely to infect projection neurons (Masamizu et al, 2011; Surdyka and Figiel, 2021). This strategy has also been used to retrogradely label midbrain DANs that project to the striatum (Cearley and Wolfe, 2007; Crittenden et al. 2016). We have added these literature references to the revised manuscript (Lines 250-251).

Reviewer #2 (Remarks to the Author):

This manuscript investigated the intrinsic properties, synaptic inputs, and threat conditioning behavioral contributions of substantia nigra pars lateralis dopaminergic neurons (SNL DANs). This study is significant because in contrast to SN pars compacta (SNC), the function of SNL DANs is less well understood. In particular, the intrinsic properties and excitatory inputs that drive SNL DAN firing are poorly characterized. The authors identified SNL DANs as a functionally distinct subpopulation characterized by irregular pacemaking and significantly higher maximal firing rates compared to SNC DANs. A key finding was that, unlike SNC DANs, SNL DANs receive robust inputs from auditory association cortices. In behavioral experiments, the authors found that disrupting synaptic release from the auditory cortex to SNL DANs impaired the retrieval of auditory threat memories, but interpreted their data as showing that threat conditioning was unaffected. For the most part this is a well-structured and well-written study that makes a novel and important contribution to the dopamine research field by elucidating the function of an under-studied dopaminergic subpopulation.

However, there is one large weakness of the study in its present form: the noisy nature of the data in Fig. 5 does not adequately support the claim that “preventing synaptic release from auditory cortex to SNL DANs did not affect threat learning”. In fact, the authors themselves

acknowledged a “clear trend towards diminished freezing behavior during conditioning” (page 7). Looking at the data in Fig. 5e, the difference seems more than a mere trend (one time point even has a star above it, which suggests statistical significance). Clarifying this issue would be important for strengthening the rigor of the study (it would have no effect on the study’s significance). This could be resolved through additional experiments to increase the sample size. If this is not possible, then the authors would need to tone down the claim that threat learning was not affected. At best, the present data suggests that threat learning is affected by a smaller amount than the retrieval phase.

We agree that comparison of the two groups led to an interpretation that was unclear. To address this, we used a more appropriate statistical test, the two-way ANOVA, to compare control and TetTx treated animals during threat conditioning. The results of our new analysis showed that TetTx treated animals freeze significantly less overall during both phases of threat conditioning (control, N = 8; TetTx, N = 6, $p=0.012$; 2-way ANOVA). Post-hoc comparison revealed that there was significant difference between control and TetTx treated mice during retrieval ($p=0.02$; Sidak’s post hoc) but not conditioning ($p=0.08$). As mentioned by the reviewer, we agree that the small sample size of this experiment limits our ability to make conclusions about threat learning during the conditioning phase alone. Therefore, we believe that the significant reduction in freezing in the TetTx treated mice during retrieval likely results from a deficit in threat learning during the conditioning phase which our analysis was not able to detect. Thus, the conclusions in our revised manuscript only support significant differences between control and TetTx mice overall, but not between conditioning and retrieval.

Consistent with this finding, we performed new threat conditioning experiments on a separate cohort of animals (see comment by Reviewer 3) and found similar findings. We found that control and TetTx treated mice show statistically significant differences overall, but do not differ statistically in their freezing during conditioning or retrieval phase. When considered together, these two independently performed sets of experiments provide strong evidence that synaptic transmission from the corticonigral pathway supports threat conditioning processes. We included a discussion section that addresses differences between learning and retrieval (Lines 305-317 and 386-395).

A moderate weakness was the lack of sufficient information regarding the statistical comparisons used to analyze the results. The p-values and type of test used should be included in the figure captions.

We have now included p-values and the type of statistical tests used in the figure captions.

Minor comment:

Typo on page 7, should be “we recorded a significant increase in % freezing...”

This has been corrected.

Reviewer #3 (Remarks to the Author):

The authors investigated the role of substantia nigra pars lateralis (SNL) dopaminergic neurons (DANs) in auditory threat conditioning. Using functional mapping, optogenetics, and slice electrophysiology, they characterized the firing properties of SNL DANs, identified excitatory inputs from the auditory association cortex to SNL DANs, and examined the functional roles of their projections to the tail of the striatum (TS). They reported that disrupting synaptic transmission in these projections impaired auditory threat learning and memory retrieval, suggesting the importance of the SNL-TS pathway in aversive learning.

To meet the standards for publication in Nature Communications, several key issues need to be addressed. Below, I outline major concerns that should be resolved to strengthen the manuscript.

1. Identification of Neurotransmitters Released by the Auditory Cortex onto SNL DANs (Figure 2): The study highlights the specific projection of the auditory cortex to SNL DANs. However, the neurotransmitters released by the auditory cortex onto DANs and the mechanisms underlying this communication remain unclear. Further investigation into the specific neurotransmitters and receptor types involved would greatly enhance our understanding of this circuitry.

Thank you for raising this point. **We have performed additional electrophysiology and optogenetics experiments to demonstrate that corticonigral projections release glutamate that acts on ionotropic glutamate receptors (AMPA/NMDA) to excite SNL DANs.** We drove excitation in SNL DANs by optically stimulating cortical terminals from the AuV/TeA which resulted in an increase in firing. We found that application of NBQX and AP5 resulted in an inhibition of optically evoked firing, suggesting that excitation from corticonigral terminals is mediated by AMPA/NMDA receptors. These new experimental data have been included in the revised version of Figure 2j,k.

2. Introduction of STN and PPN (Figure 3):

The authors introduce the subthalamic nucleus (STN) and pedunclopontine nucleus (PPN) as distinct inputs to SNC and SNL, but their functional role in auditory threat learning has not been investigated. This inclusion feels abrupt and underdeveloped. Further examination of their role would enhance the coherence of the study. Additionally, the schematic in Figure 6 should integrate the functional roles of STN and PPN to provide a more complete picture.

We agree that the experiments examining the inputs from STN and PPN were peripheral to the main observations of the study (See also comment from Reviewer 1). Because of this, we have removed all data and discussion related to STN and PPN projections from the revised manuscript.

3. TetTx Injection (Figure 4):

A critical issue lies in the authors' approach to silencing synaptic transmission from SNL to the TS. TetTx was injected into the TS using DAT-Cre mice, which silenced DAT-expressing neurons in the TS, but not in the SNL, because AAV entered into the cell bodies and expressed TetTx in the infused area. To effectively address whether disrupting the SNL-TS pathway interferes with auditory threat learning, TetTx should have been injected into the SNL via a projection-specific

manner. The current results from this experiment are therefore inconclusive, and further experimentation is needed to support these claims.

We can clarify our experimental approach here. Our approach in this experiment was to *retrogradely* infect DANs by injecting an AAV-DJ in the TS using a DAT-Cre mouse. The AAV-DJ virus serotype infect cells through the soma but, importantly, has also been shown to retrogradely infect DANs through their axonal projections to the striatum (During et al 2020). In Figure 3 of our study, we also used an AAV-DJ for retrograde infection of DAN. Specifically, we injected a Cre dependent AAV-DJ-TetTx (AAV-DJ CMV DIO eGFP-2A-TeNT) into the TS in a DAT-Cre mouse which led to expression of TetTx in the SNL DANs that project to the TS.

The reviewer also commented that our viral manipulation would silence “...*DAT-expressing neurons in the TS, but not in the SNL, because AAV entered into the cell bodies*”. We want to clarify that there are very few to no cell bodies that are DAT-expressing in the striatum (see IHC image of DAT-Cre mouse from Allen Brain Atlas Experiment #978). Thus, the TS viral injection that we performed would lead to expression of TetTx only in Cre-positive SNL DANs via retrograde infection of their axonal terminals in the TS. Therefore, the viral expression shown in Figure 3 represents a projection specific approach which exclusively targets SNL DANs that project to the TS.

IHC image from Allen Brain Atlas experiment (#978) showing a mouse coronal section with DAT+ neurons in substantia nigra (left, black). Note the lack of labeling in TS (right).

We also revised Figure 3 to provide a clearer representation of the viral strategy (see image section below).

4. Threat Conditioning Protocol (Figure 5):

While TetTx was used correctly in this figure, the threat conditioning protocol has inconsistencies. The tone durations during habituation (30 s) and conditioning (10 s train) differ significantly, which may introduce confounds. Additionally, the protocol would have been more robust if the authors had used a Context ABA design. Furthermore, the schematic in Figure 5i does not accurately reflect the described protocol and should be revised for clarity.

In response to this concern, **we have performed *new behavioral experiments that 1) use exactly the same tone duration (30s) during both conditioning and retrieval phases and 2) use a context ABA design.*** In agreement with our previous experiments, we found that inhibition of the corticonigral projections to SNL with TetTx impairs threat conditioning. We have substituted the previous version of these experiments with new data now shown in Figure 4h-l of the revised manuscript.

Fig. 4. | Auditory threat conditioning is modulated by auditory association cortex neurons projecting to SNL. **h**, Behavioral paradigm used for simple tones auditory threat conditioning. **i-k**, Graph showing average % freeze for habituation/conditioning, retrieval, and contextual phases. Squared symbols represent the freeze average during a single pure tone for habituation/conditioning and retrieval (control, N = 8; TetTx, N = 8). **l**, Average % freeze during simple tones. Every squared symbol represents average from a single mouse. Two-way ANOVA showed significant treatment ($p = 0.0398$), but not phase (0.51), or phase-treatment interaction ($p = 0.98$). Average % freeze resulted significantly different between control and treated mice for Contextual Freezing ($p = 0.048$, Mann-Whitney). All bars shown Day 1 with \pm s.e.m.

5. Minor comments:

- Any abbreviations (e.g., HCN) should be introduced where they first appear, not later in the discussion.
- The terms “DAN” and “DA neurons” are used interchangeably. It would be clearer to choose one term and use it consistently throughout the manuscript.

These have now been corrected.

Reviewer #4 (Remarks to the Author):

Thank you for your time and helpful suggestions.

We thank the reviewers for their comments and for the time spent reviewing our manuscript. We were excited to see that the reviewers were satisfied with our responses to their criticisms and that the revised manuscript has addressed their concerns.

Reviewer #1 (Remarks to the Author):

Thanks to the authors for their time in revising this paper. Overall the paper has been heavily revised and updates, including multiple new experiments and analyses. I think the paper is overall stronger and the previous comments have been addressed.

Thank you.

Reviewer #2 (Remarks to the Author):

I am satisfied with the changes the authors made to address my concern about former Fig. 5. The new results and interpretation enhance the rigor and significance of the study. I have no further concerns. The study presents interesting and novel findings on the synaptic properties and functional role of a little-studied population of dopaminergic neurons. One minor suggestion is to improve the wording of the title in Fig. 3, which is currently a bit vague.

Thank you for the suggestion and for your careful review of our manuscript. We have rephrased the title for Figure 3.

Reviewer #3 (Remarks to the Author):

All my concerns have been clearly resolved now. Thank you to the authors for the thoughtful and thorough revisions. I am happy to recommend the manuscript for acceptance.

Thank you.

Reviewer #4 (Remarks to the Author):

We thank this Early Career Researcher for their time and effort on this review.